# Current and Emerging Disaster Risks Perceptions in Oceania: Key Stakeholders Recommendations for Disaster Management and Resilience Building

**DOI:** 10.3390/ijerph16030460

**Published:** 2019-02-05

**Authors:** Joseph Cuthbertson, Jose M. Rodriguez-Llanes, Andrew Robertson, Frank Archer

**Affiliations:** 1Disaster Resilience Initiative, Accident Research Centre, Clayton, Monash University, Melbourne 3800, Australia; Frank.Archer@monash.edu; 2European Commission, Joint Research Centre (JRC), 21027 Ispra, Italy; jose-manuel.rodriguez-llanes@ec.europa.eu; 3Western Australia Department of Health, East Perth, WA 6004, Australia; Andrew.Robertson@health.wa.gov.au

**Keywords:** disaster risk, Oceania, emerging risk, health threat, resilience, non-traditional

## Abstract

Identification and profiling of current and emerging disaster risks is essential to inform effective disaster risk management practice. Without clear evidence, readiness to accept future threats is low, resulting in decreased ability to detect and anticipate these new threats. A consequential decreased strategic planning for mitigation, adaptation or response results in a lowered resilience capacity. This study aimed to investigate threats to the health and well-being of societies associated with disaster impact in Oceania. The study used a mixed methods approach to profile current and emerging disaster risks in selected countries of Oceania, including small and larger islands. Quantitative analysis of the International Disaster Database (EM-DAT) provided historical background on disaster impact in Oceania from 2000 to 2018. The profile of recorded events was analyzed to describe the current burden of disasters in the Oceania region. A total of 30 key informant interviews with practitioners, policy managers or academics in disaster management in the Oceania region provided first-hand insights into their perceptions of current and emerging threats, and identified opportunities to enhance disaster risk management practice and resilience in Oceania. Qualitative methods were used to analyze these key informant interviews. Using thematic analysis, we identified emerging disaster risk evidence from the data and explored new pathways to support decision-making on resilience building and disaster management. We characterized perceptions of the nature and type of contemporary and emerging disaster risk with potential impacts in Oceania. The study findings captured not only traditional and contemporary risks, such as climate change, but also less obvious ones, such as plastic pollution, rising inequality, uncontrolled urbanization, and food and water insecurity, which were perceived as contributors to current and/or future crises, or as crises themselves. The findings provided insights into how to improve disaster management more effectively, mainly through bottom-up approaches and education to increase risk-ownership and community action, enhanced political will, good governance practices and support of a people-centric approach.

## 1. Introduction

Disaster risk reduction is a holistic approach to long-term community and state development that unifies preparation and mitigation practices. Its function has been described as “lying between the interface of humanitarian response to disasters and developmental programs [1].” Traditional emergency management-based frameworks employ hazard and risk analysis in the development of prevention, preparedness, response and recovery plans. This analysis is invariably based on historical data of event impact. Flage and Aven explored definitions of emerging risk and described a knowledge-based definition where emerging risk is “related to an activity when the background knowledge is weak but contains indications/justified beliefs that a new type of event (new in the context of that activity) could occur in the future and potentially have severe consequences to something humans value” [2].

The scale, frequency and type of disasters are changing globally [3]. Academic and humanitarian institutions engaged in disaster-related science have identified new and emerging health threats to populations [4]. These threats have been described as underlying drivers that increase the frequency, complexity and severity of disasters. Described underlying drivers include poverty, climate change, poor governance, ecosystem decline, rapid urbanization and population growth [3]. These underlying drivers have yet to be clearly defined either in terms of scope and to what extent they aggravate the impact of a disaster. Moreover, it is unclear whether some of these aggravating factors might be considered as new or emerging, and as yet unclassified, disasters. Existing disaster databases that rely on pre-determined definitions, thresholds and data reporting systems, are not always sensitive to these changes and may not readily identify new or non-traditional health threats. The emergence of new, unexpected, non-traditional threats and disaster types, and the re-emergence of former health risks, has received less attention in the literature and requires broader and diverse approaches to the investigation of future disaster threats to population health [5]. Identifying emerging risks to health in Oceania and understanding the context of disaster risk through the perceptions of relevant stakeholders working with the at-risk population provides an opportunity to inform perceptions on emerging risks and consider appropriate disaster risk management strategies.

A comprehensive approach to reduce disaster risk was mandated in the United Nations-endorsed Sendai Framework for Disaster Risk Reduction, whose declaration was “to enhance efforts to strengthen disaster risk reduction and to reduce disaster losses of lives and assets from disasters worldwide” [6]. The Sendai Framework for Disaster Risk Reduction recognizes the need for improved assessment of emerging disaster risk with a specific call to “academia, scientific and research entities and networks to focus on the disaster risk factors and scenarios, including emerging disaster risks, in the medium and long term” [6]. Small island states are specifically identified by the Sendai Framework for action with a call for “particular attention” to their higher vulnerability and risk [6,7].

Utilizing the Sendai Framework lens of applying a “comprehensive approach” to disaster risk reduction, this study examined perceptions of current and future disaster risks in Oceania and contrasted them to reported, classified disaster impacts. Our investigation is reflective of the Sendai Framework by increasing research for regional, national and local application, supporting the interface between policy and science for decision-making and describing risk priorities [6]. We sought to delineate priority areas for action relative to emerging disaster risk. This study addresses the research question “do knowledge gaps exist between what disaster risks are measured in Oceania and what disaster risks are perceived to be of threat by experts within the region?” 

## 2. Materials and Methods

The study used a mixed methods approach. The definition of Oceania for this research was applied according to the United Nations Statistical Division methodology. Quantitative data on the natural and technological disaster impact in Oceania from 2000 to 2018 was extracted from The Centre for Research on the Epidemiology of Disasters (CRED) Emergency Events Database (EM-DAT) [8]. The EM-DAT, a free, publicly accessible web-based resource, contains essential core data on the occurrence and effects of over 22,000 mass disasters in the world from 1900 to the present day. The criteria for EM-DAT database entry of a reported disaster is ten or more people reported killed, or one hundred or more people reported affected, or declaration of a state of emergency or a call for international assistance. The EM-DAT data includes: Disaster number, country or countries in which the disaster has occurred, disaster group, disaster type, date, number of deaths, number of people injured, number of people homeless, number of people affected, and estimated damage [8].

Qualitative data was collected through key informant semi-structured interviews. Interview questions are included in Appendix A. Using convenience sampling, thirty-five interviews, contacted by email or face-to-face, were requested. Of these, 30 accepted to be interviewed while the remaining 5 did not respond to the email request. All of the participants were engaged in disaster management in the Oceania region as researchers, practitioners in emergency management or disaster healthcare, policy managers or academics and were identified through existing disaster management professional networks within the region. Thirty individual interviews with participants from 9 different countries were conducted face to face (*n* = 19) or by telephone call (*n* = 11). Data collection was conducted between April and November 2017, and a typical interview lasted between 45 and 60 minutes. Interviews were recorded and ceased when responses indicated that no new information was obtained. The Hazard and Peril Glossary for describing and categorizing disasters applied by CRED [8] guided thematic analysis of interview questions one and two (Appendix A). Comparison of individual responses to questions 1 and 2 (Appendix A) to the reported pattern of disasters in country of the respondents’ location in Oceania (Tables 2 and 3) was conducted to identify differences in perceived current and emerging disaster risks compared to reported events. Qualitative analysis of the remaining interview questions used narrative inquiry according to the six-step process described by Braun et al. [9]. Interviewee responses were analyzed using the thematic analysis where a theme “captures something important about the data in relation to the research question, and represents some level of patterned response or meaning within the dataset” [9]. Developed themes were reviewed to identify similarity or overlap and whether unification into central themes or sub themes was appropriate.

### Ethical Considerations

All respondents provided written, informed consent prior to participation, provided by scanned version or picture by e-mail, and did not receive any incentives to participate in the study. Ethical approval was obtained from Monash University Human Research Ethics Committee (HREC 7539). Recordings were stored safely and kept confidential as per The Australian Code for the Responsible Conduct of Research, and the Monash University HREC Guidelines.

## 3. Results

The demographic profile of interviewees for the analyzed sample (Table 1) showed similar proportions across gender, slightly favoring females. The response rate from Australian interviewees (60%) was large; however, interviewees from 6 countries in Oceania participated in the study. Practitioners in emergency management or disaster healthcare were the predominant profession interviewed. Three participants were located outside of Oceania (Indonesia, Geneva); however, their experience and knowledge of disaster risk reduction in the region provided a valuable contribution.

### 3.1. Reviewing Potential Themes

Initial findings from interview question one (Appendix A) were developed into a theme (1) of perceived climate change risk (Box 1). Exploration of risk within interviewee responses demonstrated varying contexts relative to the interviewees’ location in the Oceania region; this contextual risk perception was developed as theme 2. Many responses clustered around human development, human health and human impact, both internal and external to Oceania, as factors influencing disaster risk, which was unified as a singular theme (theme 3). There was inconsistent response to the utility of disaster plans, with a perceived gap between planners, actors and leaders, and themes identified in answers to question two, which recognized a lack of political leadership and poor governance associated with events of internal conflict, migration and extremism. A singular theme of planning and governance was derived from these findings (theme 4). Local education, community engagement and leadership support were identified as drivers of change and formed theme 5. The final five themes resulting from our analysis are presented in Box 1.
Box 1Final five themes identified in this study.Climate change is observed as a contemporary and emerging disaster risk in Oceania.Risk is contextual to the different countries, communities and individuals in Oceania.Human development trajectories and their impact, along with perceptions of a changing world, are viewed as drivers of current and emerging risks.Current disaster risk plans and practices are not suited to the future disaster risks: Reconnect end users (community) and developers (government). Enhanced political will and good governance are key.Increased education and education of risk and risk assessment at a local level to empower community risk ownership. leadership and action: A people-centric approach.


**Theme 1: Climate change is observed as a contemporary and emerging disaster risk in Oceania.**


Respondents across Oceania reported climate change as a top current (and future) disaster risk. How climate change was described as a risk or a hazard varied between participants. Sea-level rise was identified twice as a current risk, both times by residents of the Pacific Islands, who, along with other respondents, also reported concerns of increased sea transportation accidents due to rising sea levels and trans-national migration in island areas due to loss of habitable area [Respondent 8, New Zealand; Respondent 28, Hawaii; Respondent 6, Tonga]. Current climate change risk was also identified in association with disease “*Climate related infectious disease/pandemic*” [Respondent 11]. Overall, participants who viewed climate change as a risk or hazard described a breadth of impacts related to it, including “*Increasing global warming influencing natural disaster risk*” [Respondent 18] and “*Climate change, lack of water, heat related injuries/consequences, cyclones and natural disasters increasing in severity*” [Respondent 7]. Whereas some respondents highlighted a relationship between climate change and disaster risk, other participants described additional health threats and vulnerabilities associated with climate change. Responses included: “*Climate change—its impact on livelihood and health*” [Respondent 22], “*Climate change, lack of ability of communities to cope with impact*” [Respondent 5], “*Climate change, diseases, trans-nation migration due to climate change*” [Respondent 8, New Zealand].


**Theme 2: Risk is contextual to the different countries, communities and individuals in Oceania.**


The geography and location of Oceania is a factor related to its natural risk profile and was consistently reported by interviewees as a significant contemporary and emerging threat in Oceania. Risk is contextual to place and person in Oceania; in particular, respondents noted cyclone and earthquake events as high disaster risks in Oceania. Cyclone was identified as a current or emerging threat in 18 instances, whereas earthquake was identified as a current or emerging threat in 21 instances. An international emergency response health practitioner related natural hazards and risk in Oceania, “*the location of Oceania lends itself to these risks*” [Respondent 28, Hawaii]. Furthermore, there was a reported sense of a lack of health security to protect communities. In particular, “*Oceania neighbours have decreased levels of public health infrastructure to mitigate and/or protect against the impacts of disasters*” [Respondent 18, Australia], and they also argued that population growth is an influencing factor: “*What’s reported seems to indicate that they (disasters) are escalating in size and population numbers are increasing therefore the footprint is increasing*” [Respondent 18, Australia]. Interviewees across all countries expect natural disasters to decrease and disease events to remain relatively steady. This was an unusual finding, given that the frequency and severity of natural disasters is expected to rise and the risk profile of Oceania is high, due to the presence of tectonic fault lines and seasonal cyclone and drought patterns [10].


**Theme 3: Human development trajectories and their impact, along with perceptions of a changing world, are viewed as drivers of current and emerging risk.**


Human development trajectories and their impact are viewed as drivers of current and emerging risk in Oceania. Perceptions of a changing world are considered to be related to perceived risks. Population growth and changes in population demographics, transport, technology and human security, the effect of urbanization and the impact of human development on the environment, including biodiversity loss and pollution, were viewed as current and emerging risks in Oceania by respondents.

Perceptions of disaster risk associated with conflict were not constrained to acts of terrorism, although terrorism was viewed predominantly as a current rather than future disaster risk. Impacts of events due to conflict and crime, including cyber-attacks and technology security, criminals acting under the banner of terrorism, unexpected behavior of individuals/violence, and civil unrest, featured as current and future reported risks. Of these, one respondent noted that events such as “*terrorism and infectious disease are global/intercontinental risks that seem to be media exposed rather than public health exposed*” [Respondent 10, New Zealand].

The impact of urbanization was considered to be a current and future driver of disaster risk. In particular, food, water and energy security due to unplanned population growth or growth that exceeded infrastructure capacity, were identified. Additionally, unrealistic expectations of improved living standards were reported as urban-related hazards that increase disaster risk. Several respondents identified “*increasing demand exceeding infrastructure, unsustainable capacity to respond adequately to emerging threats, human development and its imbalance with nature, and affluent society having high expectations that may not be met post impact of a disaster*” [Respondents 12, 19] as drivers of disaster. Respondent 1 reported that “*within urban settings a sense of community has reduced over years particularly in big cities. In condensed areas there is an increased risk of disruption to basic needs*” [Respondent 1]. A new risk of plastic pollution was reported as a current and emerging disaster risk by the same respondent [Respondent 1, Australia].

Additional emerging non-traditional disaster risks associated with transport identified by respondents included fossil-fuel dependence for transport of goods, political risk from migration, transportation need increasing due to rising sea levels, road traffic incidents in Pacific Island Nations due to increased westernization of the population, and novel microbial infections in poorer nations due to frequent travel in the region.

Human development was considered to be an underpinning driver of new disasters in Oceania.


**Theme 4: Current disaster risk plans and practices are not suited to the future disaster risks: Reconnect end users (community) and developers (government). Enhanced political will and good governance are key.**


Current disaster risk plans and practices are not considered suited to the future disaster risks. Moreover, there is a perceived disconnect between the end user (community) and the developer (government). Lack of political will and poor governance are viewed as barriers to improvement. Respondents did not demonstrate a strong belief that disaster risk plans and practices are suited to the future disaster risks. Only eight of thirty respondents felt that current disaster risk plans and practices are suited to the future disaster risks. A further eight were unsure and one respondent felt that “*They are extremely variable across regions which is a major issue, more theoretical than practical*” [Respondent 5].

A lack of trust and belief in government were expressed by 14 respondents. Given the high proportion of Australian respondents, further examination of the context of trust and governance in this country is worthy of further exploration Other reports of “*government corruption, political disasters, political unrest, lack of government strategy and planning to manage this effectively, and rogue political states*” [Respondent 24, Indonesia] were used as descriptors of barriers to improvement.

Investment in activities that enhance disaster-risk reduction require long-term political vision and will to enable policy and planning that increases security at an individual, community and national level over time. Such decisions can be challenging where political systems lack strength to enable and commit to long-term sustainable development.


**Theme 5: Increased education and education of risk and risk assessment at a local level to empower community risk ownership, leadership and action: A people-centric approach.**


Community-based action was a common response as a solution to improve disaster-risk reduction. Achieving action in this area included suggestions of training and education related to understanding and owning risk at an individual and community level “*Putting the right tools and resources into local systems to better manage their own risk*” [Respondent 23]. Linkage between theme four and theme five was evident as respondents reported the need for leadership and governmental support in achieving these actions.

A people-centred approach was evident in respondent suggestions. Key responses noted areas for improvement of disaster practice that can enhance future community resilience to disaster risk. These included: “*Ensuring grassroots training on preparedness and response on the disaster risks that are relevant to those communities. Providing training to communities and ensuring plans are local and relevant; training people to self-respond, build self-awareness into communities to have initial plans at the time of need. Lessons learnt need to be applied and practiced; engaging communities, finding ways for communities to be further involved/integrated into disaster management. Minimise top down, maximise bottom up strategies. Resource communities to understand and adopt good risk assessment practices. Every community needs to own risk management strategy that is updated regularly with new and evolving knowledge. Urban planning needs disaster risk strategies built into them with detail. Then communicate these actions into the local population. Improve connectedness in communities, and knowing people and groups within them—this should be a function of disaster practice that creates trusted networks. The decision makers are still tied to response. There is a need to look at prevention with greater strength and engagement. We need to make sure that everyone within the health system has a role to play in managing disaster risk so that when ‘controls’ are overrun we transition into emergency mode. This would mean managing risk at all times—society in general does this, when vigilance drops things happen.*” [Respondent 11]. The emphasis on enabling ownership of disaster risk at an individual and community level was common amongst all respondents. The provision of guidance, information, and access to services that incorporate and build on social norms and cultural practices were suggested to improve program integration and enable informed decision-making by citizens.

Increased education of risk and risk assessment at a local level is recommended to develop downstream activities with upstream commitment and enhance community connection with risk ownership, leadership and action. Improvement requires a people-centric approach that is supported at all levels.

### 3.2. Quantitative Results

The Emergency Events Database defines Oceania according to the United Nations Statistical Division methodology. The EM-DAT reporting of natural disasters data classified according to the Hazard and Peril Glossary [11] in countries of the respondents location in Oceania in 2000–2018 was extracted from EM-DAT (Table 2 and Table 3) [8].

The EM-DAT data demonstrate storm events occurring in all countries reported, and this event was the highest natural disaster event type across the date range in all countries except New Zealand. Climate change is not included within the framework of disaster classification applied by EM-DAT; however, CRED does recognize climate change as an ‘exacerbating factor’ of classified disasters. The quantitative data of natural disasters in Oceania from EM-DAT did not indicate any change in rates of disasters considered to be associated with climate change (drought, storm, flood, extreme temperature).

Water-related transport disaster events occurred in all countries reported and were the highest transport-related disaster for all countries.

Australia reported the highest volume of natural and technological disaster events across the date range.

The findings of question one and two of the interviews were classified using the same methodology applied by EM-DAT [11]. When comparing the responses to question 1 of interviewees with the EM-DAT data, there are similarities in the natural hazard profile of storm, flood and earthquake and the technological hazard profile of transport accidents. When comparing the historical quantitative data in Table 2 and Table 3 to the qualitative data of current and future threats reported in questions one and two, there was a demonstrated knowledge gap between perceived and reported disasters.

## 4. Discussion

This research found that climate change is viewed as a contemporary and emerging disaster risk in Oceania. Reports of climate variability, transportation increasing due to rising sea levels, trans-national migration due to climate change, climate-related disasters, climate issues in island areas and loss of land mass were descriptors applied to describe perceived hazards and impacts due to climate change. These emerging risks are reflective of both the geographical location of countries in Oceania, where decreasing land mass due to rising oceans has been previously reported [12], and climate change-driven migration [13,14,15]. Climate change was perceived as an individual risk, and as an influencing factor on other risks, by many respondents. Climate change has been broadly associated with migration, conflict and health security by many authors [16,17,18,19]. The association between climate change-induced migration, and its relationship to transport accidents at sea, is a unique finding in this research and demonstrates the contextual risk of this hazard in the Pacific. Interviewees from Pacific Island Nations (Fiji, Tonga, Timor Leste) consistently identified natural hydrological hazards as current risks (storm and flooding) and the impacts of climate change as new emerging risks. This finding may be due to geographical positioning and respondents’ sense of health security related to their location and increased local health burden.

The finding of climate change as a current and future cause of disasters is consistent with the Fifth Assessment Report by the Intergovernmental Panel on Climate Change, which focussed on understanding and adapting to extreme events and disasters, predicted to become more frequent under climate change [20]. The Sendai Framework for Disaster Risk Reduction calls for “addressing climate change as one of the drivers of disaster risk” and suggests that “more dedicated action needs to be focused on tackling underlying disaster risk drivers, such as the consequences of poverty and inequality, climate change and variability” [6]. Moreover, climate change has been described by the Lancet Commission in 2009 as “the biggest global health threat of the 21st century” [21], and, as such, it represents a non-traditional, emerging threat to the health status of communities [22]. The threat that climate change poses as a disaster has been recognized and acted upon by the World Association of Disaster and Emergency Medicine (WADEM), which has issued a position statement and special report to inform and guide members and partners [14,23].

When considering future threats, our response data describing emerging risk in Oceania over the next 10 years (question 2) did not always fit the profile of traditionally reported disaster definitions and trends. A mixed pattern of threats that can act as risk factors or become disasters themselves emerged from the data. The effects of an increasingly urbanized region appear to be evident in the types of contemporary risks reported that are associated with drivers of human impact. Plastic pollution, cyber-insecurity, biodiversity loss, inequitable resource distribution, salinization, infrastructure weaknesses, chronic disease, transport accidents, food, water and energy insecurity, poverty, refugee crises and changing patterns of international migration were captured by the interviews. These findings are notable, as the relationship between increasing urbanization, human development and population growth in respect to pollution, unequal distribution of resources, environmental impact and disaster risk has been previously identified [24]. Monitoring and reporting on planetary health has emerged as a new discipline in academia supported by a topical Lancet journal dedicated to the investigation and monitoring of human impacts and the boundaries of planetary capacity to absorb and adapt to these [25]. The increase in disposable, single-use plastics has been identified as a driver of global plastic pollution requiring whole of government, industry and community action [26,27].

Direct transport accidents are currently classified by EM-DAT as technological disasters (e.g., death or injury as a direct result of transportation incident or event) [11]. In the context of transport-related risk in Oceania, respondents reported that increased road and sea transport, and the need for people to move and travel will increase transport risks. This included road traffic injuries, migration, increased disease risk due to increased ease of global travel, and mass transport accidents in isolated areas, as current and future drivers of transport disasters in Oceania. Whereas some of the reported threats are currently classified as disasters (i.e., transport accidents), other emerging threats are not visible as classified, reported events. As a consequence, many of the perceived emerging risks are not captured in traditional disaster event databases, as they are not viewed or defined as traditional disasters. These unclassified risks or non-traditional health threats are pervasive, as their measurement is not aligned with current disaster events to enable threat analysis. One interviewee reported “*whether society is able to cope with what is happening in respect to an aging population*” [Respondent 25, Australia] as a current risk in Oceania, inferring a sense of change in population and a lack of preparedness for the potential impact of the change. Conflict as a driver of disaster varied in context and included terrorism, unexpected behavior of individuals/violence, cyber-attack, internal conflict, political unrest, migration and extremism. This finding is consistent with contemporary research identifying conflict and terrorism as a public-health problem [28,29,30].

Reported unclassified risks are expected to continue into the next decade; however, the response rate of some indicated a reduction over time of some perceived threats (21 interviewees reported terrorism-related events as a current risk, 13 reported them as a future risk; 23 interviewees reported urbanization-related events as a current risk, 13 reported them as a future risk). The relationship of such events to natural disasters as either a mitigating or an enhancing effect is unknown, as is the context of these risks in small island states, where size and geography can limit resilience absorptive capacity.

Whereas disaster databases do not capture these unclassified risks or non-traditional health threats, global risk analysis reports have. The World Economic Forum Global Risk Report identified societal fracturing and a loss of trust by communities in government in 2014 [31]. In 2017, inequality has been reported as a key driver of contemporary risk [32]. This finding is reflective of theme four, “a perceived disconnect between the end user (community) and the developer (government)”. Lack of political will and poor governance are viewed as “barriers to improvement”; and the reporting of poverty by respondents as an emerging Oceanic risk.

The identification and development of theme five was based on consistent reporting of a need for community empowerment in improving disaster risk-reduction practice. This finding demonstrates a strong correlation with the World Health Organization’s Risk Reduction and Emergency Preparedness Strategy. This strategy has purposefully focused its action at the community level and has described goal of “participation that can measurably reduce future risks and losses” [33].

Oceania was reported as “the canary in the coal mine of emerging disaster risk, there is a lack of political attention to Oceania impacts from overseas countries, what’s happening in the Pacific is due to the effects of countries faraway” [Respondent 20]. In consideration of global policies, and statements on the protection of health related to disaster risk, these findings of unclassified risks or non-traditional health threats beg the question of when does an emerging risk, chronic threat or event impact exceed a threshold of social perception that transitions its context and classification as a disaster? Furthermore, is there a greater need to consider how changes in population demographics are used to inform risk assessment?

Solutions recommended by interviewees focussed on improving individual and community risk education, awareness, and ownership. Improvement in community and governmental trust is required to facilitate such action and empower communities to engage in prevention-related strategies that are contextual to their location, capacities and vulnerabilities. As stated by one respondent, “*Communities have to own it. This means a sense of self awareness is required to understand you require a measure of self-reliance*” [Respondent 20].

### Limitations

This study sought to understand the profiles of disaster risks in Oceania and the perceptions of emerging disaster risk. This study is limited in that there is not representation from all Oceanic Pacific Islands; additionally, the majority of respondents were predominantly employed in the health sector of disaster management. As a consequence, their views may be more focussed on health outcomes related to disaster impacts based on insights that are related to practical experience in disaster management in Oceania. A predominantly Australian response rate (60%), compared to all other countries accessed in Oceania, may bias the perception of overall responses to a more Australian perspective. The rates of natural disasters in Oceania considered to be related to climate change were considered in this research; however, the impact of such events on any change was not, which is a limitation of this paper.

## 5. Conclusions

Natural disasters related to the geography and location of countries in Oceania were identified by respondents as significant current and future disaster risks in Oceania. Non-traditional health threats and unclassified risks were an additional feature of our investigation. This was of particular relevance to thematic findings 1 and 3: Climate change as a contemporary and emerging disaster risk, and human development trajectories and their impact and perceptions of a changing world viewed as drivers of current and emerging risk in Oceania. Disaster reporting is typically limited to established classification and human impact outcome. However, the evolution of new and non-traditional health threats can also cause emergency situations. These forces threaten community and public health and well-being, and warrant rethinking about how one classifies disasters in the context of emerging threats to health. Such consideration has implications for the application of the Sendai Framework for Disaster Risk Reduction and activities designed to mitigate disaster impact, and may leverage functions of community resilience and sustainable development. The reported emerging disaster risks in Oceania are not captured by traditional disaster definitions and classification methodology or in existing disaster databases. To improve sensitivity of detection, it is recommended that a review of disaster classification contextual to emerging threats to health is conducted, and a monitoring program is established to identify and track these drivers of risk.

Current disaster risk management plans should be revised and enhance bottom-up approaches for risk management that develop and enable community action. Action should be initiated to implement education and ownership of risk reduction practice at a person and community level. Such action is reflective of the thematic findings 2 and 5, where engagement, education and partnership with communities in understanding risk, hazard and impact lends itself to shared responsibility. Controls of current and future risks should be developed that are sensitive to human development and the environment, and contextual to the local population.

Thematic finding 4 described in this research provides opportunities to inform, update and improve policy and practice at a regional level. Achieving action in all these areas is a great challenge as it requires long-term political vision and will to enable policy and planning that increase security at an individual, community and national level. Furthermore, if implemented, these practices must be people-centered. The provision of guidance, information, and access to services that incorporate and build on social norms and cultural practices will enhance trust, improve program integration and enable informed decision-making by citizens. Facilitation and leadership at a government level is required to guide this process to achieve activities that are designed to mitigate impact and improve resilience. In short, good governance is a cornerstone for successful implementation of disaster risk reduction strategies.

This study identified key knowledge gaps between measured and perceived disaster risks in Oceania. Recommendations based on comparison of thematic analysis of expert opinion of perceived disaster threats and measured disaster events are reported to inform improved disaster risk reduction practices.

## Figures and Tables

**Table 1 ijerph-16-00460-t001:** Participant demographics.

Characteristics	No. (%)
*Gender*	
Male	14 (46%)
Female	16 (53%)
*Country*	
Australia	18 (60%)
New Zealand	2 (6%)
Indonesia	2 (6%)
Timor Leste	2 (6%)
Fiji	2 (6%)
Tonga	2 (6%)
Hawaii	1 (3%)
Geneva	1 (3%)
*Profession*	
Academic	5 (16%)
Practitioner (emergency management or disaster healthcare)	23 (76%)
Manager	2 (6%)

**Table 2 ijerph-16-00460-t002:** Natural disasters of study countries in Oceania 2000–2018 [8].

Country/Natural Disaster	2000	2001	2002	2003	2004	2005	2006	2007	2008	2009	2010	2011	2012	2013	2014	2015	2016	2017	2018	Grand Total
Australia	6	9	5	11	7	4	8	2	5	6	8	1	2	2	2	7			1	86
Drought			1				1												1	3
Epidemic				1																1
Extreme temperature										1					1					2
Flood	1	4		4	3	1	1	1	4	2	5	1	2							29
Insect infestation	1				1															2
Storm	3	3	3	5	3	2	3	1	1	2	3				1	4				34
Wildfire	1	2	1	1		1	3			1				2		3				15
Fiji	1	1		1	2	1	2	4	1	2	1		3			1				20
Drought																1				1
Flood	1				1	1	1	2		1			2							9
Storm		1		1	1		1	2	1	1	1		1							10
New Zealand	2	1	1	2	2	2	1	1						3			2	1		18
Drought														1						1
Earthquake														1			1			2
Epidemic				1																1
Extreme temperature		1																		1
Flood	1		1		2	1	1										1			7
Storm	1			1		1		1						1				1		6
Tonga		1			1					1		1			1		3			8
Earthquake										1										1
Storm		1			1							1			1		3			7

**Table 3 ijerph-16-00460-t003:** Technological disasters in study countries in Oceania 2000–2018 [8].

Country/Natural Disaster	2000	2003	2004	2005	2007	2009	2010	2011	2012	2013	2016	Grand Total
Australia	2		1	1	1	1	1	1	1	1		10
Air				1								1
Fire	1							1				2
Rail			1		1							2
Water	1					1	1		1	1		5
New Zealand		1					2	1	1			5
Air									1			1
Explosion							1					1
Other								1				1
Water		1					1					2
Tonga						1						1
Water						1						1

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
