# Peer review of "Current and Emerging Disaster Risks Perceptions in Oceania: Key Stakeholders Recommendations for Disaster Management and Resilience Building"

_ijerph, 2019, doi:10.3390/ijerph16030460_

Reviewer 1 Report

The manuscript “Current and emerging disaster risks perception in Oceania…” presents results from 30 surveys of persons in the health sector of disaster management.  Ten questions were asked of each respondent (Appendix 1).

The authors have some interesting results but the results are presented in a rambling manner that does not allow the reader to find the main points.  Lines 108-118 refer to Box 1 (which needs an explanatory caption).  The themes mentioned in lines 108-118, such as risk perception (line 111), disaster risk (lines 112-113), and planning and governance  (line 116) do not align with Box 1.

Climate change is said several times to be the major threat but lines 136-137 state that geography and location are “perceived to be the greatest contemporary and emerging threat in Oceania.”  This is not clear.

Table 2 has too many categories of disasters for a diagram of this type.  The nations do not align clearly with the horizontal bar charts.

Table 3 lists technological disasters but it is not clear how the reader is to use the information.  What does it tell us about risk perception in Oceania.  There is internal inconsistency of the total affected population being fewer than total deaths in most cases.  Boat and plane crashes are not commonly considered technological disasters.

Plastic pollution is mentioned in the Discussion on line 235, but I did not see it in the survey results.

Lines 310-311 state that climate change induced migration was shown to be related to transport accidents but I did not see that in the results.

The Discussion is long and rambling.  It is not always clear whether the authors are stating their own findings or summarizing literature on these subjects.

Ten questions were asked of the respondents (lines 386-401).  I did not see a clear description of the answers given by the 30 respondents to those 10 questions. 

The Conclusion leads by saying that non-traditional health threats and impacts causing social disruption were a feature of this investigation (lines 364-365).  Yet the respondents mostly talked about the traditional threats of climate change (lines 234, 242), heat, sea level, disease, health, urbanization, poverty, and so on.

Overall, the results need to be presented in a more structured manner that allows the reader to understand what the 30 respondents said when asked the 10 questions.

End of comments.

Author Response

Response to Reviewer 1 Comments

Dear reviewer, many thanks for your guidance and feedback. We have reviewed the manuscript in respect to your findings and have provided responses below for your consideration.

The manuscript “Current and emerging disaster risks perception in Oceania…” presents results from 30 surveys of persons in the health sector of disaster management.  Ten questions were asked of each respondent (Appendix 1).

The authors have some interesting results but the results are presented in a rambling manner that does not allow the reader to find the main points.

Response: The results section has been revised into a structured, sequential manner of qualitative and quantitative findings (line 122-288)

Lines 108-118 refer to Box 1 (which needs an explanatory caption). 

Response: An explanatory caption has been included. (line 144)

The themes mentioned in lines 108-118, such as risk perception (line 111), disaster risk (lines 112-113), and planning and governance  (line 116) do not align with Box 1.

Response: Further detail on the derivation of the themes risk perception (line 162), disaster risk (lines 165), and planning and governance (line 222) has been included, which aligns with Box 1.

Climate change is said several times to be the major threat but lines 136-137 state that geography and location are “perceived to be the greatest contemporary and emerging threat in Oceania.”  This is not clear.

Response: The wording has been adjusted reflective of the thematic analysis where respondents across Oceania reported climate change as a top current (and future) disaster risk. (line 146)

Table 2 has too many categories of disasters for a diagram of this type.  The nations do not align clearly with the horizontal bar charts.

Response: The table design has been revised to enhance readability.

Table 3 lists technological disasters but it is not clear how the reader is to use the information.  What does it tell us about risk perception in Oceania.  There is internal inconsistency of the total affected population being fewer than total deaths in most cases.  Boat and plane crashes are not commonly considered technological disasters.

Response: The table has been revised with explanatory notes and findings (line 281-287)

Plastic pollution is mentioned in the Discussion on line 235, but I did not see it in the survey results.

Response: The interview results of the qualitative research that include plastic pollution have been included in the results section (line 203)

Lines 310-311 state that climate change induced migration was shown to be related to transport accidents but I did not see that in the results.

Response: The results of the qualitative research that include climate change induced migration shown to be related to transport accidents have been included in the results section (line 205-209)

The Discussion is long and rambling.  It is not always clear whether the authors are stating their own findings or summarizing literature on these subjects.

Response: The discussion has been revised to explore result findings and relative literature (line 288-380)

Ten questions were asked of the respondents (lines 386-401).  I did not see a clear description of the answers given by the 30 respondents to those 10 questions.

Response: The answers were thematically analysed to derive thematic findings, the thematic findings are referred to in the results section. The manuscript has been revised to include relevant quotes and detail to support the author’s thematic findings.

The Conclusion leads by saying that non-traditional health threats and impacts causing social disruption were a feature of this investigation (lines 364-365).  Yet the respondents mostly talked about the traditional threats of climate change (lines 234, 242), heat, sea level, disease, health, urbanization, poverty, and so on.

Response: The conclusion has been revised to provide more concise recommendations related to thematic findings (line 393-428)

Overall, the results need to be presented in a more structured manner that allows the reader to understand what the 30 respondents said when asked the 10 questions.

Response: The answers were thematically analysed to derive thematic findings, the thematic findings are referred to in the results section. The manuscript has been revised to include relevant quotes and detail to support the author’s thematic findings. (line 145-262)

End of comments.

Reviewer 2 Report

Topic is likely to be of interest, and is a significant topic.

Strengths:
The article brings attention to a critical subject through a new lens;
Qualitative approach is a strength;
Article also highlights policy considerations reflecting a particularly vulnerable region;

Weakness:
“Whilst” is a word that would be considered atypical English to a readership in North America and perhaps should be replaced by the word “while.”

As noted by authors, the preponderance of interviews occur with Australian respondents as opposed to others in the Pacific region.

Author Response

Response to Reviewer 2 Comments

Dear reviewer, many thanks for your guidance and feedback. We have reviewed the manuscript in respect to your findings and have provided responses below for your consideration.

Topic is likely to be of interest, and is a significant topic.

Response: Thank you for your support

Strengths:

The article brings attention to a critical subject through a new lens;

Qualitative approach is a strength;

Article also highlights policy considerations reflecting a particularly vulnerable region;

Response: Thank you for your support

Weakness:

“Whilst” is a word that would be considered atypical English to a readership in North America and perhaps should be replaced by the word “while.”

Response: The manuscript has been updated to include this change (line 156, 338, 355)

As noted by authors, the preponderance of interviews occur with Australian respondents as opposed to others in the Pacific region.

Response: We have ensured this is clearly stated in the manuscript limitations (line 382-391)

Reviewer 3 Report

The paper approach is a very interesting one and well described in chapter 2. However abstract and introduction are less clear and would need a bit of restructuring.

in addition the paper promises an analysis of Oceania while the vast majority of interviewees comes from Australia, raising questions on how far results can be evaluated and conclusions can be drawn regarding Oceania as a whole. The authors stress that in their limitations, however pointing at the origin of sources e.g. when giving quotes would be recommendable. Interviews results are compared against EMDAT data, but the comparison remains a bit vague, the points the authors want to make in the discussion chapter are not fully clear.In addition, giving selected quotes should be backed by further information related to the other interviewees - does the selected quote reflect their opinion, are there differences/similarities, e.g. among the country/professional backgrounds? The potential of the high number of interviews does not seem to be tapped fully.

The authors try to assess/elaborate on health risks from a DRM/DRR perspective, however there is hardly any framework/reference/approach from the DRR side mentioned.

The conclusions partly don't refer to the text but rather bring up new/rarely mentioned aspects

- "Non-traditional health threats": this is not linked to the disaster context and hardly defined, thus it remains a very blurry term

- "emerging disaster risks in Oceania are not captured by traditional disaster definitions and classification methodology or in existing disaster databases": there is no reflection on definitions throughout the paper, the same is true for methodologies and databases with the exception of EMDAT

minor comments

- I disagree to the first sentence "Emerging disaster risks are poorly understood" - there is huge and growing body of literature on that topic, so please either give a reference/elaborate or modify

- 18:why to improve the disaster management cycle? rather the activities within different steps or the understanding of the cycle? unclear

- 24: why only emerging and not anyway existing risks? what exactly does that cover? - contradicts with 27 where some risks that are not always linked to disaster are mentioned, thus it remains unclear how the authors define risk in this paper

- 39-42: this is a strong statement definitely lacking references. what kind of new risks?

- table 2: title wrong, the disasters mentioned are not only "natural"

- table 3: may be included in table 2?

- 347-352: very brief, there surely are more recommendations?

Author Response

Response to Reviewer 3 Comments

Dear reviewer, many thanks for your guidance and feedback. We have reviewed the manuscript in respect to your findings and have provided responses below for your consideration.

The paper approach is a very interesting one and well described in chapter 2. However abstract and introduction are less clear and would need a bit of restructuring.

Response: We have revised the abstract and introduction accordingly to provide a focussed summary of the manuscript

in addition the paper promises an analysis of Oceania while the vast majority of interviewees comes from Australia, raising questions on how far results can be evaluated and conclusions can be drawn regarding Oceania as a whole. The authors stress that in their limitations, however pointing at the origin of sources e.g. when giving quotes would be recommendable. Interviews results are compared against EMDAT data, but the comparison remains a bit vague, the points the authors want to make in the discussion chapter are not fully clear.In addition, giving selected quotes should be backed by further information related to the other interviewees - does the selected quote reflect their opinion, are there differences/similarities, e.g. among the country/professional backgrounds? The potential of the high number of interviews does not seem to be tapped fully.

Response: Where appropriate and relevant to the finding the origin of sources and comparison has been included. EMDAT data comparison has been reduced to provide a more refined analysis.

The authors try to assess/elaborate on health risks from a DRM/DRR perspective, however there is hardly any framework/reference/approach from the DRR side mentioned.

Response: The revision provides reference to the Sendai framework (line74, 402)

The conclusions partly don't refer to the text but rather bring up new/rarely mentioned aspects

- "Non-traditional health threats": this is not linked to the disaster context and hardly defined, thus it remains a very blurry term

Response: The conclusions have been reworded to ensure they are clearly related to the results. Non traditional health threats and its relationship to the disaster context has been more clearly discussed. (line 393-428)

- "emerging disaster risks in Oceania are not captured by traditional disaster definitions and classification methodology or in existing disaster databases": there is no reflection on definitions throughout the paper, the same is true for methodologies and databases with the exception of EMDAT

Response: A definition of emerging risk has been included (line 44)

minor comments

- I disagree to the first sentence "Emerging disaster risks are poorly understood" - there is huge and growing body of literature on that topic, so please either give a reference/elaborate or modify

Response: The first sentence has been re worded emphasising the importance of identification and understanding emerging risk (line 12)

- 18:why to improve the disaster management cycle? rather the activities within different steps or the understanding of the cycle? unclear

Response: The sentence has been reworded (line 33) to improve disaster risk management practice

- 24: why only emerging and not anyway existing risks? what exactly does that cover? - contradicts with 27 where some risks that are not always linked to disaster are mentioned, thus it remains unclear how the authors define risk in this paper

Response: The sentences have been reworded (line 12) to capture current and future disaster risks.

- 39-42: this is a strong statement definitely lacking references. what kind of new risks?

Response: The sentence has been reworded and references have been included (line 49-53)

- table 2: title wrong, the disasters mentioned are not only "natural"

Response: The table title is reflective of events drawn from EMDAT

- table 3: may be included in table 2?

Response: The table has been kept separate to delineate the impacts of transport related technological disasters that are relevant to Oceania and the perceived threats from respondents

- 347-352: very brief, there surely are more recommendations?

Response: The recommendations have been revised related to the thematic findings (line393-428)

Round  2

Reviewer 1 Report

Authors have adequately addressed my concerns.